# Preparation and Evaluation of PDMS/Carbon Soot Particles Superhydrophobic Biomimetic Composite Coating with Self-Cleaning and Durability

**DOI:** 10.3390/biomimetics7030132

**Published:** 2022-09-13

**Authors:** Fengqin Li, Yong Liu, Honggen Zhou, Guizhong Tian

**Affiliations:** College of Mechanical Engineering, Jiangsu Provincial Key Laboratory of Advanced Manufacturing for Marine Mechanical Equipment, Jiangsu University of Science and Technology, Zhenjiang 212100, China

**Keywords:** superhydrophobicity, candle soot, biomimetic composite coating, self-cleaning

## Abstract

In this paper, a superhydrophobic biomimetic composite coating was fabricated on brass by electrochemical etching, brushing PDMS adhesive layer, and depositing carbon soot particles. Due to the microstructure and the optimized ratio of PDMS, the contact angle of the superhydrophobic coating is up to 164° and the sliding angle is only 5°. The results of optical microscopy and morphometric laser confocal microscopy show that the prepared coating surface has a rough hierarchical structure. A high-speed digital camera recorded the droplet bouncing process on the surface of the superhydrophobic coating. The self-cleaning property of the coatings was evaluated by applying chalk dust particles as simulated solid contaminants and different kinds of liquids (including grape juice, beer, cola, and blue ink) as liquid contaminants. The coating remained superhydrophobic after physical and chemical damage tests. This work presents a strategy for fabricating superhydrophobic biomimetic composite coatings with significant self-cleaning properties, durability, and shows great potential for practical engineering applications.

## 1. Introduction

Brass is an important engineering material and is widely used in the chemical and marine industries due to its excellent thermal stability, good electrical conductivity and, outstanding corrosion resistance [1,2] The preparation of superhydrophobic coatings based on the brass substrate is one of the key research directions in the field of surface engineering [3,4,5]. A superhydrophobic surface is a surface with a contact angle greater than 150° and a sliding angle less than 10° [6]. The artificial superhydrophobic surface was first inspired by the hydrophobic and self-cleaning phenomena on the surface of the stems and leaves of some plants in nature, as well as on the feathers or skin surfaces of some birds and animals. For example, the lotus leaf always keeps its clean leaf surface because of its unique superhydrophobic property. According to a previous study [7], the upper epidermis of the lotus leaf has micrometer-scale rough papillae structures surrounded by hydrophobic wax-like substances. Benefiting from the special papillary structure and the hydrophobic wax-like substance, the two basic conditions for realizing superhydrophobic function, namely micro-nano structure, and low surface energy, are satisfied. (Literature support) When the water droplets roll off the upper surface of the lotus leaf, it can take away the pollutants on the leaf surface, so as to achieve the effect of self-cleaning, which is the famous “lotus leaf effect” [8]. At present, the applications of superhydrophobic coatings include digital microfluidics [9], oil-water separation, [10] fluid drag reduction [11], droplet manipulation [12], anti-fouling [13], anti-icing [14], and metal corrosion protection [15], etc. And it is evident that the research of superhydrophobic coatings with excellent performance has good prospects. To achieve a superhydrophobic surface, two principles need to be satisfied. One is to obtain a rough hierarchical structured surface, and the other is that the material has low surface energy.

According to these two principles, researchers have two directions for preparing superhydrophobic surfaces: either to reduce the surface energy of the rough surface or take measures to make the material surface with low surface energy [16].

To achieve the required coating, researchers have developed various strategies in the preparation process. In terms of techniques, electrostatic spinning [17], chemical vapor deposition [18], electrochemical deposition [19], and laser ablation [20] are commonly used. In terms of materials, researchers chose to add used nanoparticles such as SiO_2_, ZnO, and TiO_2_ to provide roughness or fluorides and non-polar molecules to provide low surface energy [21]. Although these strategies have made some progress, there are still many problems, such as the expensive equipment and materials involved. Faced with these problems, carbon soot particles, as a cheap hydrophobic material with high porosity and high stratification dimension, are easily accessible and low cost [22]. However, the direct use of carbon fume particles to prepare superhydrophobic surfaces without any treatment is unsatisfactory. Because there is only a weak physical force between carbon nanoparticles, they are very fragile. They can be easily peeled off from the substrate surface by external mechanical forces, water erosion, etc. The use of binders or crosslinkers can significantly improve the shortcomings of the superhydrophobic character based on carbon soot particles and enhance the adhesion and bonding of carbon soot particles on the surface of the substrate [23]. Seo et al. [24] used paraffin wax to fix fragile candle soot. Compared with soot coating without any treatment, the robustness and durability of the coating have been significantly improved. However, the author himself also pointed out that the mechanical stability of paraffin is poor, and it is easy to scratch. Paraffin is also decomposed due to thermal degradation when the temperature exceeds 250 °C. Xiao et al. [25] connected carbon particles through the hydrolysis of methyl trichlorosilane (MTCS) in the air. While the coating showed excellent resistance to corrosion in acidic and weakly alkaline liquids as well, it was less effective against strong alkaline solutions. Moreover, the chemical vapor deposition method involved in this study has a slow deposition rate and the gaseous by-products are usually very toxic. These drawbacks limit this study to large-scale commercialization. The polymers used in their study by Sutar et al. [26] require solvents such as toluene, acetone, and o-xylene to dissolve, and the rate of substrate immersion and withdrawal in the polymer solution has to be controlled by the dip coater. These commercial polymer materials and equipment devices undoubtedly indirectly increase the cost of actual industrial production. To overcome the above shortcomings, relatively simple electrochemical etching and direct deposition methods have been adopted by us.

Here, superhydrophobic coatings were prepared on brass surfaces by electrochemical etching, brushing the PDMS layer, and depositing carbon soot particles. First of all, the surface of the brass substrate was electrochemically etched, which makes the substrate surface have an uneven microstructure. Subsequently, a 5:1 ratio of PDMS prepolymer and curing agent was used as a binder to coat the etched brass substrate. Then, hydrophobic soot particles were deposited directly in the middle of the candle flame to prepare the superhydrophobic coating. The surface micromorphology and wettability of the prepared superhydrophobic coatings were characterized and investigated, and the self-cleaning properties of the coatings were confirmed by solid and liquid contaminant tests. In addition, corresponding physical and chemical damage tests, including water jets, water droplet impacts, sandpaper abrasion, and acid and alkaline liquid immersion, were designed to further evaluate and verify the durability of the coatings, and the relevant properties of the coatings were enhanced thanks to the embedded and semi-embedded structures formed between the PDMS adhesive layer and the carbon soot particles, and the experiments showed that the prepared coatings can be applied for a long time in real environments.

## 2. Experimental Details

### 2.1. Materials

Commercially available brass plates (2 cm × 2 cm × 0.25 cm) were used as substrates. Copper sulfate was used as a raw material for the electrolyte solution. It was procured from Sinopharm Chemical Reagent Co. Candles (1.5 cm × 17 cm) were procured from the local supermarket and were used as a source to produce carbon soot particles. PDMS (Sylgard 184) was purchased from Dow Corning (Midland, TX, USA). Absolute ethanol, citric acid, and sodium carbonate were purchased from Aladdin Reagent Company (Shanghai, China).

### 2.2. Preparation of Superhydrophobic Surfaces

The preparation process of superhydrophobic coating mainly consists of electrochemical etching, PDMS layer, and carbon soot particle deposition. Firstly, the brass substrate was sanded and polished with 800-grit sandpaper to remove the oxide layer on the surface, and then ultrasonically cleaned with anhydrous ethanol for 5 min. The brass substrate was etched in copper sulfate solution (concentration of 20 g/L) at an optimized current density of 2 A/dm^2^. The brass substrate was used as the anode and the stainless-steel sheet was used as the cathode. The distance between the anode and cathode was 30 mm. And the etching solution was stirred for 30 min at room temperature of 25 °C with a magnetic stirrer of 1000 r/min. After etching, the brass substrate was washed with anhydrous ethanol for 5 min and then dried in an oven at 60 °C for 10 min. Subsequently, the PDMS mixture (weight ratio of prepolymer to curing agent = 5:1), was brushed uniformly onto the etched brass substrate. In order to allow PDMS molecules to penetrate into the substrate better, the etched brass substrate after brushing was placed in a vacuum drying oven with a pressure of 0.06 MPa for 20 min.

During the deposition process, the pretreated brass substrate was placed in the middle flame position of the candle and hydrophobic carbon soot particles (CSP) were directly deposited for 2 min. And the superhydrophobic coating (SC) on the brass substrate was obtained. The schematic diagram of the complete preparation process of the superhydrophobic coating samples is shown in Figure 1.

### 2.3. Characterization Methods

#### 2.3.1. Morphological Characterization

An optical microscope (Olympus BX53, Beijing, China), a laser confocal morphometric microscope (VK-X1000, Shanghai, China), and a scanning electron microscope (Zeiss, Jena, Germany) were used to collect the surface microscopic morphology of the sample.

When observing with an optical microscope (Olympus BX53), the sample to be observed was placed on a carrier table so that the surface of the sample to be observed was located directly below the through-hole. The sample is first observed with a low magnification lens, and then further observed with a high magnification objective lens.

The laser confocal morphometric microscope (VK-X1000) adopts a pinhole confocal optical system, and the light receiving element is a 16-bit induction photomultiplier and ultra-fine color CMOS. According to the material, shape, and measurement range of the sample, functions such as automatic upper and lower limit setting function, high-speed light quantity optimization function, and insufficient reflected light quantity compensation were selected, and high-precision measurement was carried out. The magnification of the objective lens is 20 and 50 times, the wavelength of the laser used for measurement is 408 nm, the maximum output power is 0.9 mw, and the display resolution is 0.5 nm.

The accelerating voltage (EHT) of the scanning electron microscope was 5 kV, the working distance (WD) was 7.8 mm, and the detector (Signal A) used was SE2.

#### 2.3.2. Contact Angle and Sliding Angle Measurement

The contact angle values of the sample surfaces were determined by optical contact angle meter (Theta Lite, Shanghai, China) measurements. The volume of water droplets used for the measurements was 5 μL, and the contact angle measurements were performed at three different locations on the surface of each sample and their average values were calculated. A high-speed camera (PCO.dimax S1, Munich, Germany) was used to observe the droplet state on the surface of the sample in real-time, and the tilt angle of the sample was controlled by the auto-lifter at a rate of 1 cm per minute. The camera captured the instantaneous image of the droplet when it started to roll down along the surface, and then measured the angle of the droplet when it rolled down from the surface of the sample in the instantaneous image using DWRuler software, which is the sliding angle.

#### 2.3.3. Self-Cleaning Test

To evaluate the self-cleaning performance of the sample pieces, chalk ash particles and different common household liquids (including grape juice, beer, cola, blue ink, and milk) were used as the simulated contaminants used for the test. In this study, large diameter chalk ash particles were ground with a grinder, and then the ground chalk ash was screened with a 200-mesh sieve, at which point multiple sample pieces were placed under the sieve to ensure that these samples were uniformly sprinkled with chalk ash of the same particle size at the same moment.

#### 2.3.4. Physical and Chemical Damage Tests

The mechanical durability of the sample parts was investigated by sandpaper abrasion, water jet impact and water drop impact tests.

Sandpaper abrasion is the most widely used method to assess the mechanical robustness of coatings. In this study, the sandpaper abrasion test was designed using the ISO 8251-2018 standard, and the back side of the sample was glued to the bottom of a 100 g weight with double-sided tape, and the coated side was laminated to the surface of a 1500 mesh sandpaper. The surface of the sample was effectively abraded by moving 20 cm in a straight line.

In this study, a 10 mL syringe was filled with water with the needle at an angle of 45 degrees to the horizontal. Then, the water in the syringe was pushed out rapidly and the water jet impacted the surface of the sample.

The water droplet impact was tested by allowing the water droplets to fall onto the coating from a height of 20 cm. The sample was tilted at 45° and held in place. The number of drops falling was determined from the total volume of water drops collected. The weight of 10 drops of water was measured (0.101 g/10 drops) and the volume of the drops was approximately 10 μL.

In addition, the acid and alkali resistance of the sample was examined by immersing the sample in solutions with different pH values (pH = 2, 4, 6, 8, 10, and 12) for 12 h.

## 3. Results and Discussion

### 3.1. Surface Morphology and Wettability

In this paper, the preparation of super-hydrophobic coating on brass surfaces mainly included electrochemical etching and direct deposition of soot particles. The surface morphology and wettability of the coatings were characterized using optical microscopy, scanning Electron microscope, morphometry microscopy, and contact angle meters. The test results of the bare substrate sample (BS), etched substrate sample (ES), and superhydrophobic coating sample (SC) are shown in Figure 2.

The first column of Figure 2 shows the morphological characteristics and surface contact angles of samples. As can be seen from Figure 2(a1), the BS surface has only scratches left by sandpaper sanding, and the surface contact angle caused by scratches is 107°, which is shown in the inset of Figure 2(a1). Similarly, these microstructures on the surface of ES can also be observed in Figure 2(b1), where the gully-like microstructures created air pockets beneath the water droplets. These air pockets may cause the contact angle of interfacial water droplets to increase to 130°shown in the inset of Figure 2(b1). BS and ES exhibit a degree of hydrophobicity, probably due to the adsorption of organic compounds from ambient air on the surface of BS and ES, which are mainly non-polar C-C/C-H groups. As pointed out in the literature [27], the absorption of organic matter from the surrounding atmosphere can cause a conversion of wettability. The synergistic effect of the non-polar groups and the rough surface structure make BS and ES hydrophobic.

The contact angle of the SC is 164° shown in Figure 2(c1) inset, which indicates that the SC surface is superhydrophobic. The second column of Figure 2 shows the surface microscopic images of the three specimens: the BS surface is flat, the ES surface has uneven pits due to electrochemical etching, and the SC surface has a distinct peak and valley shape after the deposition of carbon soot particles. The third column of Figure 3 shows the results of the surface roughness analysis of the specimens, respectively. It can be found that the highest values of arithmetic mean deviation of contour Ra and microscopic unevenness decimal height Rz of ES are 4.836 μm and 22.600 μm, respectively, due to electrochemical etching, indicating that it has the roughest surface. In contrast, the smallest value of the contour cell mean width RSm of SC is 4.792 μm, reflecting the higher density of deposited carbon soot particles on the surface of SC.

In order to present further preparation details and results, and to analyze the reasons for the superhydrophobicity, adhesion, and stability of the coating, the pretreatment etching process is presented in more detail. Figure 3 shows the schematic diagram of BS sample electrochemical etching preparation and etching result.

Figure 3(a1,a2) display the process of producing ES sample by electrochemical etching. Figure 3(a3) shows the results after etching, and it can be seen that there are obvious gully-like microstructures on the surface of ES. The non-smooth microstructure feature can not only trap some air, create the condition for the formation of superhydrophobic character, but also provide a place for the deposition of soot particles. After obtaining ES as shown in Figure 3 above, the PDMS-coated ES was further placed in the middle of the incomplete burning candle flame by direct deposition method to capture hydrophobic soot particles. Since the brushed PDMS mixture is a flat and thin layer, part of the PDMS mixture will penetrate into the etched substrate and will be instantly solidified when burned by the flame, and at the same time, the loose carbon soot particles can be tightly anchored to the surface of the substrate, and the carbon soot particles can completely cover the PDMS surface in the form of embedded or semi-embedded, so there would not be a violent PDMS burning phenomenon.

The preparation schematic diagram, morphology, and hydrophobic phenomenon of superhydrophobic coating samples (SC) are shown in Figure 4 below. The mass ratio of the prepolymer and the curing agent in the PDMS mixture is 5:1. Increasing the ratio of the curing agent can improve the hardness and firmness of the PDMS after being solidified [28].

When depositing carbon soot particles, the PDMS mixture on the substrate surface was heat-solidified while locking the carbon soot particles firmly on the substrate surface. The carbon soot particles on the coating surface can be divided into three main categories according to the relationship between the deposited carbon soot particles and the PDMS adhesive layer.

(i) loose carbon soot particles, which are located at the top of the coating and are not in contact with PDMS but are only connected to each other by weak van der Waals forces and can be easily worn off.

(ii) Carbon soot particles partially embedded in PDMS, which improve the mechanical stability of the coating through the embedded effect with PDMS.

(iii) Completely embedded carbon soot particles, which are integrally embedded in the PDMS adhesive layer as the PDMS is solidified by high temperature.

In addition, the microstructure formed on the substrate surface under electrochemical etching can further provide a buffer and shelter for this semi-embedded and embedded structure of PDMS-carbon soot particles to improve the mechanical wear resistance of the coating [29].

Figure 4(a1) shows a schematic of ES deposited carbon soot particles. The carbon soot particles are solid by-products of the incomplete combustion of hydrocarbons in the air. Due to the presence of C-H groups, including CH_2_ and CH_3_ groups, the carbon soot particles in the inner flame show obvious hydrophobicity [30]. The formation mechanisms of carbon soot particles include uniform nucleation of carbon soot particles, growth and oxidation of carbon soot particles, coalescence of carbon soot particles, and agglomeration of carbon soot particles [31]. By weak van der Waals forces, soot particles were bound together, forming an irregular network of particles [32]. In Figure 4(a2), rough and distributed porous micromorphology were clearly observed on the coating surface. Due to the particular rough, porous structure, air can be trapped in it. So, the contact area between the coating and water droplets is reduced which contributes to the superhydrophobicity of the surface. The structure can form a physical barrier due to the possibility of creating air pockets to reduce the contact area between the substrate and the aqueous medium. Thus an evident silver mirror phenomenon can be observed (Figure 4(a3)) [33].

Superhydrophobic surfaces can be explained by Cassie-Baxter theoretical model: A large number of air capsules on the SC surface reduce the contact area of the liquid-solid interface, and the contact angle of water droplets on the gas-liquid-solid three-phase composite surface can be explained by Cassie-Baxter equation [34]:(1)$$\cos\theta_{\gamma }=f_{1}\cos\theta_{1}+f_{2}\cos\theta_{2}$$
where $\theta_{\gamma }$ is the apparent contact angle; $f_{1}$ and $f_{2}$ are the fraction of unit apparent area occupied by solid surface and air, respectively ($f_{1}+f_{2}=1$); $\theta_{1}$ and $\theta_{2}$ are the intrinsic contact angles of water droplets on a solid surface and in the air, respectively.

Here, the contact angle of air to water is $\theta_{2}$ = 180°. Thus, Equation (1) can be further simplified to the following form:(2)$$\cos\theta_{\gamma }=f_{1}\left(\cos\theta_{1}+1\right)-1$$

According to the experimental measurement results (shown in Figure 2(b1,c1)), the values of $\theta_{1}$ and $\theta_{\gamma }$ are 130° and 164°, respectively. Therefore, $f_{1}$ is calculated to be 0.1084. This means that 89.16% of the area is water droplets in contact with air.

The ability of the SC surface to repel water droplets is powerful, which can be visually observed from the above static silver mirror phenomenon. Considering that most of the practical applications of superhydrophobic surfaces are under dynamic conditions, it is essential to verify the dynamic water repellence of the coating. Here, in order to further illustrate the low surface energy and wettability of the prepared superhydrophobic coating, an experiment of water droplets bouncing against the superhydrophobic coating was conducted to evaluate the water-repelling performance of the coating from a dynamic perspective. The process of water droplet bouncing was recorded with a high-speed camera, and the specific water droplet bouncing phenomenon is shown in Appendix A.

As shown in Figure 5(a1–a8), just like rain droplets bouncing on lotus leaves in nature, the impacting droplets first spread out on the surface to their maximum diameter, then they contract to a certain extent and eventually bounce off the surface. The water droplets underwent the motion process of spreading, shrinking, bouncing back, and rolling down on the prepared superhydrophobic coating. Due to the rough and distributed porous micromorphology on the SC surface, it can trap air near its surface and form air pockets, resulting in a stable Cassie-Baxter composite contact interface [35]. The water droplets cannot exclude the trapped air when they touch the SC surface. During the water droplet spreading phase (Figure 5(a3)), a sub-stable Cassie-Baxter contact state is formed because of the pressure inside the water droplet. In the subsequent shrinkage phase (Figure 5(a4)), a recovery from the sub-stable state to the stable Cassie-Baxter contact state is achieved. The air captured the external atmospheric pressure during the droplet rebound balances (Figure 5(a5)), thus reducing the viscous resistance of the droplet during its motion on the superhydrophobic surface. This allows the droplet to rebound and roll off from the surface (Figure 5(a6–a8)). According to the above analysis, the rebound phenomenon of the impacting droplet on the surface of the SC reflects the excellent dynamic water repellency of the prepared coating.

### 3.2. Self-Cleaning Property

On the surface of the superhydrophobic coating, rolling water droplets can carry away pollutant particles from the surface, to achieve the self-cleaning effect. Here, chalk dust particles (the main components are calcium carbonate and calcium sulfate) were used to simulate solid contaminants to examine the self-cleaning effect of superhydrophobic coatings. The large-size chalk ash particles were first ground by the grinder and subsequently screened with a 200-mesh sieve to ensure that the chalk ash particles all remained around 75 μm in diameter. The results are shown in Figure 6 below.

As shown in Figure 6(a1,b1,c1), BS, ES, and SC were placed on the edge of the glass culture dish at an inclination angle of 10° (the critical value of the sliding angle of the superhydrophobic coating surface) and sprinkled with a layer of chalk dust particles. Then, water drops were dripped on the contaminated surfaces of each sample (Figure 6(a2,b2,c2), Appendix A), respectively. As shown in Figure 6(a3), the contaminant particles adhered to the BS surface. Similarly, there is still a clear collection of contaminants on the surface of ES after the water flow (Figure 6(b3)). In contrast, the droplets on the SC surface appear spherical and the contaminant particles were carried away by the rolling water droplets (Figure 6(c3)). The motion states of droplets on the surface of the samples can be divided into three types: sticking, sliding, and rolling [36,37]. Sticking and sliding droplets have a weaker ability to remove solid contaminant particles on the samples. As pointed out in the literature [38], the pore size of the superhydrophobic surface determines the lower size limit of the contaminant. The self-cleaning property of a superhydrophobic surface can be achieved as long as the particle size exceeds the pore size of the superhydrophobic surface, or the thickness of the contaminant is lower than the protrusion height of the superhydrophobic surface. For SC, since the size of the solid contaminant particles is larger than the size of the porous microstructure on the surface of the SC, the contaminant particles only contact the top of the porous microstructure, resulting in a small actual contact area. The contact form between chalk dust and the superhydrophobic surface is point contact with little surface adhesion, which means that the high capillary force caused by water droplets is higher than the adhesion force between chalk dust particles and the superhydrophobic surface, making the particles easily carried away by water droplets. In contrast, the contact area of the BS or ES surface with the contaminants is much larger, and the water droplets leave the substrate surface in a sliding or sticky manner and do not carry away a large number of contaminants from the surface, most of the contaminant particles are simply redistributed by the action of the water droplets. The above is the self-cleaning of the hydrophobic coating on solid contaminants, and the following is a test on the self-cleaning of liquid contaminants. Here, grape juice, beer, cola, blue ink, and milk were used as simulated liquid contaminants. The result is shown in Figure 7.

The self-cleaning property of the coating was examined by comparing the accumulation of liquid contaminants on the surface of the superhydrophobic coating before and after immersion in common household liquid solutions. As shown in Figure 7, the superhydrophobic coating was clean before immersion in liquid contaminants. Subsequently, the superhydrophobic coating was removed after being immersed in different liquid contaminants and the surface of the coating was still clean. Appendix A contains more details. As described in the video, the results are similar to those shown in reference [39], and the coating also performs well against contaminated liquids.

### 3.3. Durability

Superhydrophobic coatings are often exposed to physical and chemical damage when put into practical applications, thus affecting the lifetime of the coatings. Considering the actual application environment, physical damage tests such as simulated rainwater erosion and sandpaper abrasion, and chemical damage tests such as acid and alkali resistance were designed in this paper to evaluate the durability of the prepared coatings. The test process and results are shown in Figure 8.

In order to simulate the impact of natural rainwater on the coating, water jet, and water drop impact tests were set up. As shown in Figure 8(a1), the syringe was used to draw 10 mL of water, and then the tip of the needle was aimed at the coating and tilted at an angle of 45 to the plane, and the water inside the syringe was pushed out. The water jet test video can be seen in Appendix A. It can be observed from the video that the water jet was repelled by the coating surface and bounced out in the opposite direction due to the water-repellent property of the coating. In general, superhydrophobic coated surfaces will bounce back immediately when impacted by a water jet. This is because the air cushion formed on the superhydrophobic surface blocks the water jet from entering the structure of the surface [40]. In Figure 8(a2), due to the high instantaneous water pressure of the water jet, some loose carbon soot particles on the surface are washed away, which is the normal phenomenon, and the coating area position appears cratered and eroded by the water jet. The average contact angle measured in this position was 159° (inset of Figure 8(a2)), which confirmed that the coating could resist the transient erosion by the water jet. The water droplet impact experiment is shown in Figure 8(b1). Specific parameters were set as follows: 20 cm distance between the funnel lower end and platform, 45° platform inclination angle, 10 μL water droplet volume, and the speed of 50 drops/min. The sample was fixed on the inclined platform with double-sided tape. The morphology of the sample after the impact test is shown in Figure 8(b2). After 12 h of continuous water drop impact, most of the carbon soot particles on the coating surface still resided on the substrate surface. Due to the impact of water droplets, some of the carbon soot particles on the coating surface were compacted and aggregated by physical forces which led to a slight decrease in contact angle of about 153° (inset of Figure 8(b2)). The results showed that the coating has better durability compared to the bare carbon soot coating.

In order to evaluate the wear resistance of the prepared coating, a sandpaper abrasion test was designed based on the ISO 8251-2018 standard (Figure 8(c1)). Usually, the sandpaper chosen is not less than 1000 mesh, and the larger the mesh of the sandpaper, the smaller the particle size on top of the sandpaper, then the easier it is to destroy the microscopic rough structure of the coating surface, which helps to improve the accuracy of the superhydrophobic coating wear resistance test. Here we choose 1500 grit sandpaper for the test. First, the coating surface was contacted with the upper surface of sandpaper, and then 100 g weight was placed on the base. External thrust was applied to the base to simulate the rubbing and damage effect of external mechanical force on the coating in the natural environment. In the wear test, the single wear stroke is set as 20 cm. Through experiments, it was found that after six wear cycles, some scratches appeared on the surface of the coating, and loose soot particles on the surface were worn away, but most soot particles still adhered to the substrate stably. The morphology of the worn coating is shown in Figure 8(c2). The contact angle was 154°, which indicated the robust entanglement of the soot particles with the adhesive molecules. The good physical wear resistance of the prepared superhydrophobic coatings is attributed to the surface microstructure produced by electrochemical etching and the overall structure formed between the PDMS mixture and the carbon soot particles. On the one hand, the surface microstructure of the substrate produced by the electrochemical etching provides a buffer and a shelter for the coating material. On the other hand, PDMS acts as an effective binder, which combines with the carbon soot particles to form a strong and systematic monolith.

Metal alloys, when exposed to acidic and alkaline environments, are subject to corrosion by acidic and alkaline substances, which can damage the structure and strength of the metal. The superhydrophobic coating can effectively prevent the contact between the water stream containing acid and alkali ions and the metal substrate, so that the substrate can be effectively protected from corrosion. Here, citric acid and sodium carbonate were used to prepare acidic solutions with pH values of 2, 4, and 6, and alkaline solutions with pH values of 8, 10, 12, and 14, respectively. In this paper, samples were submerged in solutions with pH values ranging from 2 to 14 (Figure 8(d1)) to test the resistance of the coating to strong acids and bases.

As shown in Figure 8(d2), the coating still maintained a contact angle greater than 150° compared with the coating before the test, and the super-hydrophobicity did not disappear. The above tests show that the prepared superhydrophobic coating can survive in this acidic and alkaline environment and possesses excellent chemical durability.

## 4. Conclusions

In summary, superhydrophobic coatings with good self-cleaning and durability were prepared by coating the etched brass substrate with optimized proportions of PDMS and then depositing soot particles. The superhydrophobic coating shows a high contact angle of 164° and the low slip angle of 5°. The synergistic effect of microstructure and soot particle deposition makes the superhydrophobic coating self-cleaning, which has a noticeable effect on artificial solid dust pollutants and common liquid pollutants. In addition, the superhydrophobic coating has maintained durability in physical and chemical tests such as sandpaper wear, water flow erosion, and acid and alkaline solutions. Therefore, this can provide a direct way to develop self-cleaning and durability superhydrophobic coatings on engineering metal substrates.

## Figures and Tables

**Figure 1 biomimetics-07-00132-f001:**
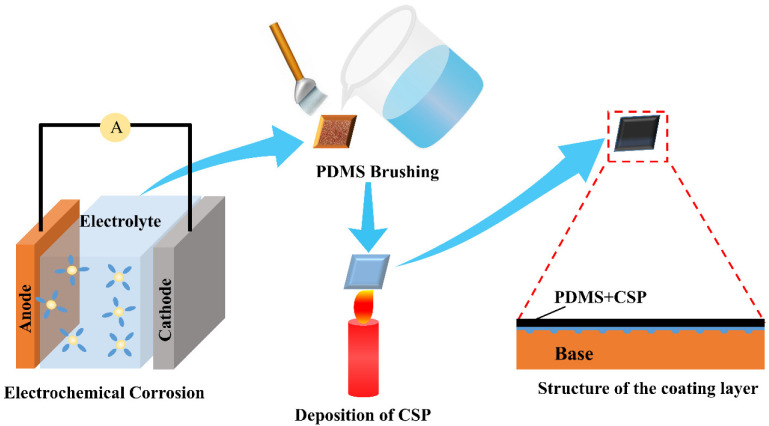
Schematic diagram of the preparation process of superhydrophobic coating samples.

**Figure 2 biomimetics-07-00132-f002:**
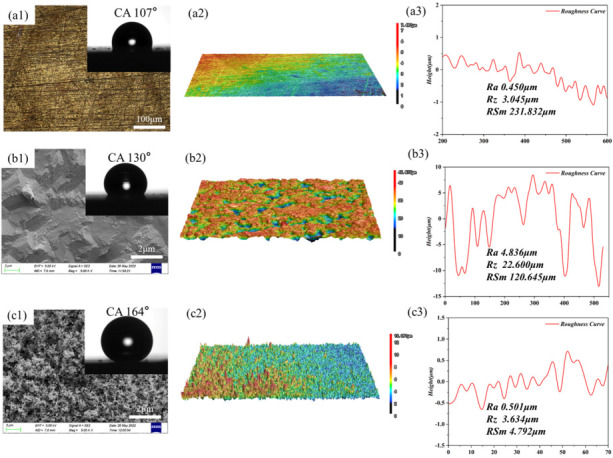
Surface morphology, contact angles, and surface roughness of samples. The first column shows the surface morphology of BS, ES, and SC, respectively; the second column shows the surface roughness of corresponding samples. The insets in the first column show the surface contact angles of each sample. The third column shows the results of the analysis of the surface roughness of the samples respectively.

**Figure 3 biomimetics-07-00132-f003:**
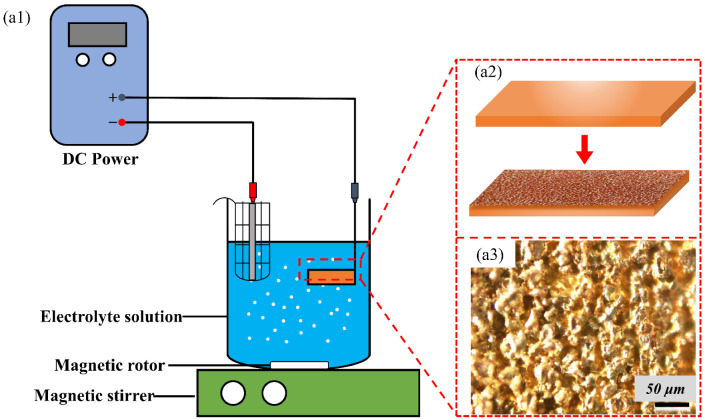
Constructing microstructures on the brass substrate by electrochemical etching: (**a1**) the schematic diagram of electrochemical etching operation; (**a2**) generation of disordered microstructures on the bare substrate by etching; (**a3**) microscopic image of the brass substrate after electrochemical etching.

**Figure 4 biomimetics-07-00132-f004:**
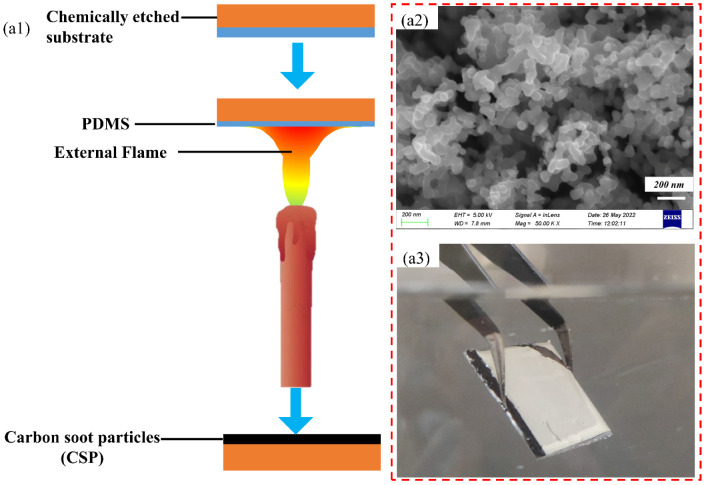
Preparation schematic diagram and the result of SC: (**a1**) schematic diagram of deposited carbon soot particles by depositing hydrophobic carbon soot particles on the etched substrate; (**a2**) microscopic morphology of superhydrophobic coating surface taken by Scanning Electron Micrscope; (**a3**) image of silver mirror phenomenon of superhydrophobic coating.

**Figure 5 biomimetics-07-00132-f005:**
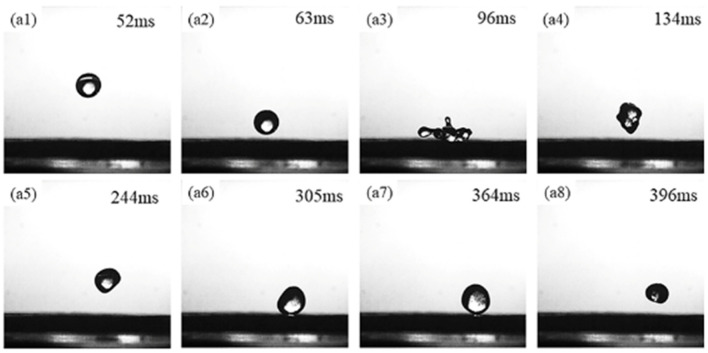
Delayed images of the water droplet bouncing phenomenon taken on the prepared superhydrophobic coating.

**Figure 6 biomimetics-07-00132-f006:**
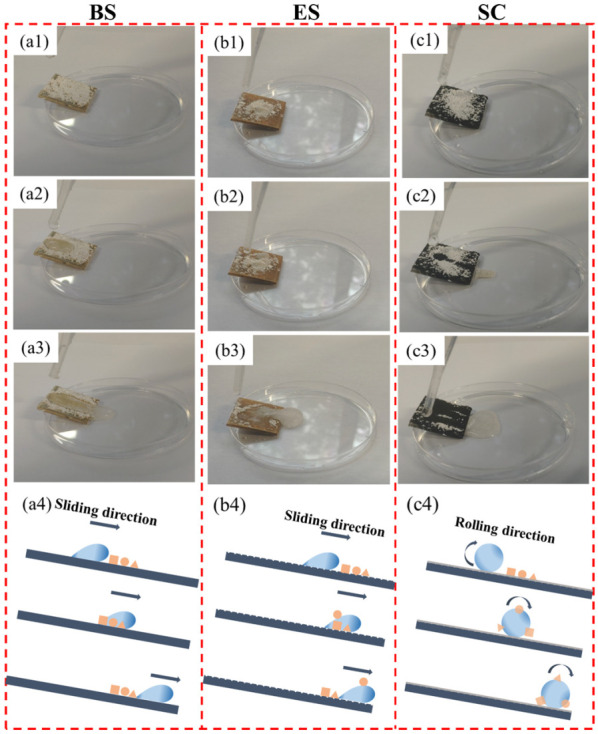
Self-cleaning tests and schematic diagrams on BS, ES, and SC against solid contaminants: (**a1**–**a3**) test process on BS; (**b1**–**b3**) test process on ES; (**c1**–**c3**) test process on SC; (**a4**,**b4**,**c4**) show the schematic diagrams of the interaction mode with the water droplets and the contaminant particles on BS, ES, and SC, respectively.

**Figure 7 biomimetics-07-00132-f007:**
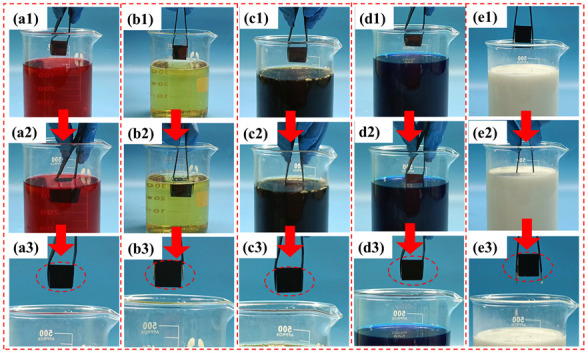
Self-cleaning tests of superhydrophobic coatings against common liquid contaminants in life: (**a1**–**a3**) test procedure in grape juice; (**b1**–**b3**) test procedure in beer; (**c1**–**c3**) test procedure in cola; (**d1**–**d3**) test procedure in blue ink; (**e1**–**e3**) test procedure in milk.

**Figure 8 biomimetics-07-00132-f008:**
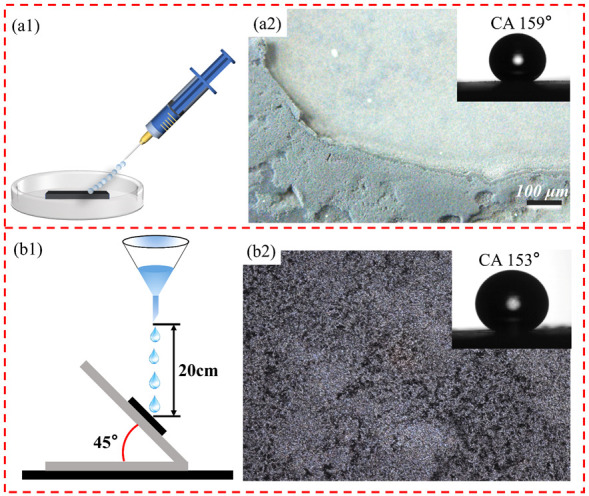
Physical and chemical damage test of superhydrophobic coating: (**a1**,**b1**,**c1**,**d1**) test process schematic diagrams of the water jet, water drop impact, sandpaper abrasion, and solution immersion respectively; (**a2**,**b2**,**c2**) microscopic topography by the microscope after samples testing, the inset is the measured contact angle; (**d2**) the contact angle after immersing in a solution of different pH for 12 h.

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
