# Peer review of "Preparation and Evaluation of PDMS/Carbon Soot Particles Superhydrophobic Biomimetic Composite Coating with Self-Cleaning and Durability"

_biomimetics, 2022, doi:10.3390/biomimetics7030132_

Round 1
Reviewer 1 Report
Li et al reported a method of preparing superhydrophobic coating on brass surface by electrochemical etching, brushing PDMS layer and depositing soot particles. The surface morphology and wettability of the prepared coating were characterized, the self-cleaning performance of the coating was tested and confirmed by artificial pollutants, and the durability of the coating under harsh conditions was verified by physical and chemical damage tests. This superhydrophobic coating can be expected to be applied in popular engineering fields such as biomedicine, photovoltaic modules, and the marine industry. In conclusion, the data in this manuscript is overall. While some explanations based on the experimental results are a little simple. Therefore, this paper can be accepted after the author address the following issues.
Comments:
1. In the introduction part, the comparison between the existing work and the previous work should be described in more detail, especially in the aspect of the mechanical stability of different superhydrophobic surfaces.
2. We all know that the vulnerability of superhydrophobic surfaces is a challenge, and some descriptions about the importance of robustness should be supplied explicitly in the introduction part.
3. In the 181 lines to 186 lines on page 6 of the sentence “the PDMS mixture on the substrate surface was heat-cured while locking the carbon soot particles firmly on the substrate surface. The bonding phenomenon occurred between carbon soot particles and PDMS molecules, forming a tangled network structure. At the same time, the surface microstructure of the etched brass substrate can withstand more pressure and act as a shelter for carbon fume particles and PDMS to resist wear and tear.” Some necessary explanations, data , or reference support should be added.
4. The 262 line on Page 6 mentions that “the solid contaminant particles is larger than the size of the porous microstructure”, If the contaminant particles are smaller than the pore size, will they be embedded in the structure? Although I don't think so, it can increase the relevant self-cleaning of small particle size pollutants to prove it. In addition, I think the hydrophilicity of pollutants also has a certain impact.
5. In the mechanical test, 6 sand-paper cycles are slightly insufficient. What if 50 cycles are performed?
6. As far as I know, superhydrophobic coatings may fail after being immersed in water or acid for a long time. The paper only shows 12 hours. What if it takes longer? How long will it expire?
Author Response
Thank you very much for your constructive comments. The specific response can be seen in the submitted word document.

Reviewer 2 Report
I have given my comments and suggestions in the attached Word file.

Author Response

(The authors gave the same response as above.)

Reviewer 3 Report
In this paper, the authors have described the synthesis procedure and characterization of superhydrophobic PDMS-coated etched brass substrates with deposited soot particles. This kind of work is definitely within the scope of the Biomimetics journal. I applaud the efforts of the authors in creating logical and extensive experiments in their work, making sure to focus on the durability of the coatings’ superhydrophobicity. The simplicity of the synthesis processes, the controls chosen and the captured videos are notable. Barring a few grammatical errors, I recommend the paper for publication after minor revisions:
1. In the introduction section, it would be useful to the reader to understand how this study relates to biomimetics (lotus leaves etc)
2. There are inconsistencies in the way peer-reviewed literature has been referenced. In some instances, the names of the authors are stated, whereas in others there is no name at all, or referred to as <first author> et.al. The latter would be preferable.
3. Considering that a candle is held to the coating to create the soot deposition, it would be pertinent to mention the flame retardance properties of PDMS – would the coating burn?
4. It would be helpful to evaluate thickness of the PDMS coating and the soot deposition. This becomes particularly important while evaluating the effect of water erosion, sandpaper abrasion and chemical corrosion. It seems there is definitely a loose layer of soot that easily gets removed, but the coating still maintains its hydrophobicity after a specific level of harsh treatments. Certain time periods like 12 hrs or number of cycles like 6 were chosen; monitoring of thickness would help other researchers extrapolate how long it would take for the soot layer to completely get removed.
5. The characterization methods section could be more detailed and divided into sub-sections.
6. The section describing the effect of several common household liquid contaminants is noteworthy but not quite helpful. It is difficult to see whether there is any sticking of those liquids on a black substrate. Perhaps contact angle measurements, or side view videos (similar to the water jet experiment, but at a slower speed) would help show if the substrate cleaned itself even after immersion in sugary and carb-heavy liquids.
Author Response

(The authors gave the same response as above.)

Round 2
Reviewer 2 Report
I applaud the diligent efforts taken by the authors to improve the manuscript following the comments given by all the referees. In the revised version of the manuscript, the presentation of the work has been improved significantly. In particular, figures have become significantly clearer, and the self-cleaning test on liquids shown in Figure 7 is much better presented now. The text has also been improved, although there is probably still some room for polishing in the level of English language. Most of the experimental details that were missing have been incorporated in the Methods section, and the interpretation of some of the tests that I found problematic before makes more sense to me now.
That said, the revisions made do not address the more fundamental issues I brought up regarding to the experimental rigor presented in the work, and thus I still find myself doubting some of the claims made regarding to the performance of these coatings. In particular, the mechanical durability of the coating is still very much in question.
Below, I will list the most significant points of criticism I still have.
- The authors have largely re-written the Introductory section, which now emphasizes the natural background of superhydropobicity as a phenomenon. While this makes sense as a framing for the journal Biomimetics, it does not follow that the coating method itself can be considered biomimetic or nature-inspired.
- Although the contact angle of the bare PDMS film (before the deposition of the carbon soot) was presented in the cover letter, it seems the information has not been incorporated in the manuscript. Furthermore, there's still no estimate given on the roughness and the thickness of the PDMS film. This is very basic background characterization information on the coating method, and should be presented even if we assume that it doesn't play a role in the performance of the coating (which I think it well might).
- Something that I didn't notice back when making my original referee statement: I think there's some confusion on the paragaph discussing the roughness measurements [LINES 252-257]: it's claimed that the ES surface has higher maximum roughness than the SC surface, and the text refers to 'Fig. 2(b3)' that doesn't exist - I assume it should be 'Fig. 2(c2)'? Also, are the subfigures b2 and c2 swapped with one another? Seeing ES surface with higher magnification now, it looks more like the surface profile in c2, and vice versa for c1 and b2. By the way, is there some reason why maximum roughness is reported as the roughness parameter? Wouldn't mean and root mean square roughnesses be more representative of the surface at large?
- The authors have made some effort to elaborate their perception on the depth-wise structure of the coatings, and suggest that the carbon soot particles form a sort of three-phase structure where particles are either fully or partially embedded in the PDMS and strongly bound to it, or loosely deposited on the top where they can be abraded away. However, no characterization data is shown to support this picture, and thus it's not much more than an assumption. I'm disappointed that the authors didn't take cross-sectional SEM pictures along the edge of the coating (with and without the soot particles) to make an effort to investigate its structure.
- For the static contact angle measurements, the authors did not include standard deviation values for the measurements on each surface. This is the bare minimum that I would expect to get a picture of the wetting uniformity of these surfaces, even if the number of measurements made was small.
- Based on the additional details given on the self-cleaning test of solid particles (chalk dust), it looks like the method did not impose control on exactly how much dust was placed on any given sample. Although I do believe that such loose debris can probably be washed away from the composite coating surface with ease, this does not qualify as a quantitative test of the self-cleaning property of the surface.
- Finally, for the sandpaper abrasion test, the authors argue that the point was primarily to prove that PDMS helps to promote abrasion resistance as compared to bare soot particle coating, and that's why additional wear cycles weren't carried out. I agree that the abrasion resistance of the surface probably benefits from the presence of a binder material such as PDMS - but that, in itself, could be a very incremental improvement. The paper does not gauge to what extent the composite coating is abrasion-resistant, and doesn't inquire how PDMS stabilizes it. The key question is whether the PDMS truly binds the soot particles in a way that is able to resist long-term abrasion, or does it simply act as a sacrificial layer in which case the coating's ability to retain superhydrophobicity is fully defined by how deep the carbon soot particles are able to migrate into the PDMS film.
To conclude, these issues mean that the technological impact of the coating method presented in this paper is still questionable. I am unsure as to whether this should mean the paper should be left unpublished in its current form, and in the end of the day that decision lies with the Editor.
Author Response
Thank you very much to the reviewer for your comments. The specific responses can be found in the uploaded word document.
